# Effects of Different TiO_2_ Nanoparticles Concentrations on the Physical and Antibacterial Activities of Chitosan-Based Coating Film

**DOI:** 10.3390/nano10071365

**Published:** 2020-07-13

**Authors:** Yage Xing, Xuanlin Li, Xunlian Guo, Wenxiu Li, Jianwen Chen, Qian Liu, Qinglian Xu, Qin Wang, Hua Yang, Yuru Shui, Xiufang Bi

**Affiliations:** 1Key Laboratory of Grain and Oil Engineering and Food Safety of Sichuan Province, College of Food and Bioengineering, Xihua University, Chengdu 610039, China; lxl0519@126.com (X.L.); gxl412326@163.com (X.G.); 18408248463@163.com (W.L.); chenjianwen0907@163.com (J.C.); xllb0519@163.com (Q.L.); xuqinglian01@163.com (Q.X.); yang1hua1@yeah.net (H.Y.); 13648022884@163.com (Y.S.); bxf1221@163.com (X.B.); 2Department of Food Science and Engineering, College of Landscape Architecture, Shangqiu University, Shangqiu 476000, China; 3Department of Nutrition and Food Science, Maryland University, College Park, MD 20742, USA; wangqin@umd.edu; 4Key Laboratory of Food Non-Thermal Technology, Engineering Technology Research Center of Food Non-Thermal, Yibin Xihua University Research Institute, Yibin 644004, China

**Keywords:** TiO_2_, antimicrobial activity, physicochemical characterization, chitosan-based coating/film

## Abstract

In this investigation, the effect of different concentrations of titanium dioxide (TiO_2_) nanoparticles (NPs) on the structure and antimicrobial activity of chitosan-based coating films was examined. Analysis using scanning electron microscopy (SEM) and atomic force microscopy (AFM) revealed that the modified TiO_2_ NPs were successfully dispersed into the chitosan matrix, and that the roughness of the chitosan-TiO_2_ nanocomposites were significantly reduced. Moreover, X-ray diffraction (XRD) and Fourier transform infrared spectroscopy (FTIR) analyses indicated that the chitosan interacted with TiO_2_ NPs and possessed good compatibility, while a thermogravimetric analysis (TGA) of the thermal properties showed that the chitosan-TiO_2_ nanocomposites with 0.05% TiO_2_ NPs concentration had the best thermal stability. The chitosan-TiO_2_ nanocomposite exhibited an inhibitory effect on the growth of *Escherichia coli* and *Staphylococcus aureus*. This antimicrobial activity of the chitosan-TiO_2_ nanocomposites had an inhibition zone ranging from 9.86 ± 0.90 to 13.55 ± 0.35 (mm). These results, therefore, indicate that chitosan-based coating films incorporated with TiO_2_ NPs might become a potential packaging system for prolonging the shelf-life of fruits and vegetables.

## 1. Introduction

Food safety has always been a concern for public. Food preservation addresses this concern in the food industry, especially in the area of ready-to-eat vegetables and fruits. During the preservation process, residual deleterious microorganisms on the surface of vegetables and fruits can propagate rapidly, due to the ethylene and carbon dioxide released by the produce during storage [1]. These may alter the quality of the fruits and vegetables, accelerate aging and rotting, causing serious economic loss, therefore, delaying the development of food industry and even endangering human’s health and life [2]. Although synthetic fungicides are effective against pathogens on postharvest fruit and vegetables, there is a greater concern about the harmful effects of fungicide residues on human health and the environment [2,3]. Therefore, it is crucial to find functional materials to effectively inhibit microbial growth and extend the shelf life of produce.

Chitosan (poly-*β*-(1→4)*N*-acetyl-D-glucosamine) is a natural macromolecule polysaccharide, which has broad applications in the preservation of fruit and vegetables, due to its properties of film-forming [4], biocompatibility [5], low toxicity [6], antimicrobial activity [7]. Chitosan is generally recognized as safe (GRAS) as a food additive by the US Food and Drug Administration (FDA) [8]. Some research results have indicated that chitosan coating can reduce postharvest diseases on a lot of produce, including apple [9], jujube [10], strawberry [11], sweet potato [12] and cherry tomato [3]. However, the bactericidal, moisturizing, mechanical and antioxidant properties of pure chitosan film have been thought unsatisfactory in practical applications [13,14]. To overcome these disadvantages, the inorganic nanoparticles (NPs) (such as silicon dioxide, zinc oxide and titanium dioxide (TiO_2_)) [15,16,17] were added during film formation, to form chitosan-based composite films, which could increase the physicochemical and biological properties.

Among the metal oxides, TiO_2_ has been found to be promising, due to its photocatalytic activity, chemical stability, low cost, biocompatibility and antimicrobial capability [8,18,19]. TiO_2_ has also been approved by the US FDA for use in human food, drugs and as a compound for food contact materials [20]. When being illuminated with UV-A light of wavelengths less than 385 nm, TiO_2_ will generate reactive oxygen species (ROS), such as ·OH, H_2_O_2_, and O_2_^−^, which are capable of destroying microbial cells and killing microorganisms [8,21,22]. Recently, much attention has been focused on the combined effect of TiO_2_ NPs in chitosan coatings on the properties of nanocomposite films, including their mechanical strength, swelling properties and thermal stability [23,24]. TiO_2_ photocatalysis has also attracted attention as a material for photocatalytic sterilization of other food and human pathogens in the food and environmental industry. When exposed to sunlight or ultraviolet light, TiO_2_ exhibits antimicrobial activity, due to its strong oxidizing properties.

In recent years, the development of composite materials made by mixing organic matter and nano-inorganic materials have obtained more attention than traditional synthetic fungicides. Several researchers have reported the effects of NPs on the properties of films. Haldorai and Shim investigated the photocatalytic and antibacterial activity of a chitosan-encapsulated TiO_2_ nanohybrid, as evidenced by the total degradation of methylene blue dye and *E. coli* within 24 h of treatment [25]. According to Li et al. [26], a nanopacking material synthesized by blending polyethylene with nano-powder (Ag NPs, kaolin, anatase TiO_2_, rutile TiO_2_), was applied effectively for the preservation of Chinese jujube to expand its shelf life and improve preservation quality. Zhang et al. [8] found that TiO_2_ nanopowders were successfully and uniformly dispersed into a chitosan matrix. Moreover, the addition of TiO_2_ led to an enhanced hydrophilicity and improved mechanical properties of the composite film. Thus, it is expected that the antibacterial activity and stability of chitosan can be enhanced by the incorporation of TiO_2_ NPs. However, to the best of our knowledge, there are no reports on the effects of different TiO_2_ concentrations on the physicochemical and antimicrobial properties of chitosan nanocomposites.

Therefore, the objective of this study was to explore the feasibility of producing antibacterial chitosan-based coating film by incorporating TiO_2_ NPs. The physicochemical properties of chitosan-based coating after its incorporation with various concentrations of TiO_2_ were investigated by scanning electron microscopy (SEM), atomic force microscopy (AFM), X-ray diffraction (XRD), Fourier transform infrared spectroscopy (FTIR) and thermogravimetric analysis (TGA) techniques. In addition, the antimicrobial activity of the newly synthesized chitosan-TiO_2_ nanocomposite was tested against two bacterial (*Escherichia coli* and *Staphylococcus aureus*) species.

## 2. Materials and Methods

### 2.1. Materials

Chitosan (85.61% of deacetylation degree) used for experiment was purchased from Jinan Haidebei Bioengineering Co., Ltd. (Shandong, China). TiO_2_ NPs (anatase-phase crystal structure with a 30 nm particle size) were purchased from Beijing Deke Daojin Science and Technology Co., Ltd. (Beijing, China). *E. coli* strain CGMCC1.0090 and *S. aureus* strain CGMCC1.8721, stored at the Institute of Fruit and Vegetable Preservation and Processing of Xihua University (Sichuan, China), were provided by the China General Microbiological Culture Collection Center (CGMCC, Beijing, China). All other chemicals were of analytical grade, unless stated otherwise. Deionized water was used in the experiments, and glassware for experiment on microbiology was autoclaved at 121 °C for 20 min.

### 2.2. Preparation of Surface-Modified TiO_2_ NPs

TiO_2_ NPs (5 g) was gently dispersed in 70 mL deionized water. The suspension was adjusted to pH 5 with 1.0 M HCl and 1.0 M NaOH, and 0.75 g sodium laurate was added (sodium laurate could expose non-polar groups of TiO_2_ NPs and disperse them better). The mixture was magnetically stirred at 40 °C for 30 min, after which the modified TiO_2_ NPs were washed with distilled water three times through a centrifuge at rate of 12,100× *g* (centrifuged for 8 min each time), and the supernatant was discarded. The modified NPs were dried at 105 °C [24].

### 2.3. Preparation of Chitosan-Based Coating Film with Modified TiO_2_ NPs

Chitosan films and chitosan-based nanocomposite coating films were prepared according to the method of Rhim et al. [27]. For preparing the chitosan film, chitosan powder (1 g) was dissolved in a mixture of 1% (*v/v*) acetic acid aqueous solution (100 mL) and glycerin (1.0 g) stirred constantly with a magnetic stirrer and heated for 20 min at 90 °C. The purpose of adding glycerin is to improve the mechanical properties of the composite film. The dissolved solution was strained through eight layers of cheesecloth to remove undissolved debris and then sonicated for 30 min in a bath-type ultrasound sonicator. The membrane solution (15 mL) was casted onto a plastic plate (d = 90 mm) and the films were dried at 25 °C for 72 h, before being peeled off.

Chitosan-based nanocomposite films were prepared by adding different concentrations of TiO_2_ NPs. First, the various concentrations of modified TiO_2_ NPs (0 g, 0.01 g, 0.03 g, 0.05 g, 0.07 g and 0.09 g) were dispersed in a 1% acetic acid solution (100 mL) containing 1.0 g glycerin. The solutions were mixed well, then sonicated for 10 min in a bath-type ultrasound sonicator to obtain a NPs solution. Chitosan powder (1 g) was then dissolved into the solution and the mixture was heated for 20 min at 90 °C under stirring. Finally, the solution was put through filtration, ultrasound, casting, drying and peeling, following the same procedure as described in the preparation of chitosan film.

### 2.4. SEM and AFM Analysis

Morphology was characterized by SEM. Different treatments, including the chitosan-based films with different concentrations of TiO_2_ NPs and blank film were placed on the stainless steel stage using double-sided adhesive tape and SEM analyses were conducted using a JSM-7500F SEM (JEOL, Beijing, China) at a voltage of 10 kV acceleration after Pt sputtering. Morphological observations of the TiO_2_ NPs powder were conducted before and after modification, in order to understand changes in its dispersion properties. The samples of TiO_2_ powder were characterized by the SEM for morphology analyses at a voltage of 15 kV acceleration after Pt sputtering [26,28].

The coating films with different concentration of TiO_2_ NPs were observed by AFM, according to the method reported by Xing et al. [29], Xing et al. [30] and Zdunek and Kurenda [31]. First, the chitosan coating films were cut into thin pieces (10 mm × 10 mm) using a small sharp knife and stuck on the stage. Ten pieces of film per treatment were scanned using the Tapping mode of a QScope250 AFM (Quesant Instrument Corporation, Agoura Hills, CA, USA). In order to understand the roughness values (i.e., Ra and Rq), the images were analyzed with a Nanoscope software (Version 5.12, Tokyo, Japan).

### 2.5. Thermogravimetric Analysis, X-Ray and FTIR Characterizations

Thermogravimetric analysis was carried out using a TGA/DSC 2/1600 analyzer (Mettler-Toledo, Switzerland). Before analyzing, the equipment was calibrated with calcium oxalate as a standard reference. Samples were placed in alumina pans and heated from 30 °C to 800 °C at a rate of 10 °C/min, under a dynamic synthetic N_2_ atmosphere at 50 mL/min [32]. X-ray diffractograms were obtained at room temperature using a Panalytical Empyrean X-ray diffractometer (PANalytical B.V., Holland) equipped with a Cu Kα1 operating at 35 kV and 30 mA. XRD patterns were recorded in an angular range of 5° to 100° (2θ), with a step of 0.026°. Fourier transform infrared (FTIR) spectra of the films were measured with a Nicolet 6700 (ThermoFisher, Waltham, MA, USA) in the reflectance spectrum mode. The spectra were obtained at resolution 4 cm^−1^, averaging over 32 scans in the range of 650 to 4000 cm^−1^ [33].

### 2.6. Determination of Inhibitory Zones against E. coli and S. aureus

Antibacterial activity of the chitosan-TiO_2_ coating films against *E. coli* and *S. aureus* was determined using an Oxford cup method, with some modifications, as described by Li et al. [34] and Karthikeyan et al. [16]. In a bacteria-free environment, stainless steel tubes (6 mm inner diameter) were placed on a nutrient agar plate pre-spread with 100 µL of microbial cell suspension (10^6^–10^7^ CFU/mL). Then, 100 μL of chitosan-TiO_2_ solution was added to each stainless steel tube, using a sterile pipette, and 100 µL of chitosan solution was included as a negative control. The plates were incubated at 37 °C for 24 h in the dark, and the zones of inhibition were measured with a caliper. Experiments were carried out in triplicate.

### 2.7. Statistical Analysis

Experimental data was analyzed by SPSS 21.0 software (SPSS Inc., Chicago, IL, USA) and reported as the mean ± S.D. The analysis of AFM for chitosan coating film was performed in duplicate (10 pieces per treatment). Other treatments were conducted in triplicate for each. The significant differences among the treatments were determined by one-way analysis of variance (ANOVA), followed by the Student-Newman-Keuls test at *p* < 0.05. Graphics analyses were conducted using Origin 9.0 (Origin Lab Co., Boulder, CO, USA).

## 3. Results and Discussion

### 3.1. Morphological Observation by SEM

The morphology of chitosan-TiO_2_ nanocomposites observed by SEM was beneficial in evaluating the effects of the composite synthesis process. Figure 1 shows the images of chitosan-based coating film with TiO_2_ NPs. As can be seen from Figure 1a, the chitosan film without TiO_2_ NPs showed light yellow color. The film color had a significant change with the increasing concentration of TiO_2_ NPs. When TiO_2_ NPs concentration increased to 0.09%, the film became white. Figure 2a shows the SEM image of the original TiO_2_ NPs received from the supplier, which apparently existed in the form of agglomerates. The morphology of the TiO_2_ NPs was, however, significantly different after surface modification, as shown in Figure 2b. The original TiO_2_ NPs existed in the form of agglomerates, however, the agglomeration phenomenon was obviously weakened, and the TiO_2_ NPs showed good dispersion after they are modified. Furthermore, in order to verify if the TiO_2_ NPs have been incorporated into chitosan-based coating film, the morphology of the chitosan coating with or without the TiO_2_ NPs was also characterized. SEM images illustrate the morphology of chitosan-based coating films with different concentrations of TiO_2_ NPs (Figure 3a–f). As indicated by Figure 3a, the surface of chitosan coating is smooth, and no cracks are found. The addition of TiO_2_ NPs has changed the microstructure of the composite coating significantly. In Figure 3b–f, it can clearly be seen that the chitosan-TiO_2_ nanocomposites show uneven nanocomposite clusters, with rough surfaces and spherical primary particles. This proves that chitosan and TiO_2_ NPs were well mixed together.

The morphology of microcapsules containing TiO_2_ NPs might be affected by the combined function of chitosan as a complex material, with less permeability for mass external and internal environments. It has been shown in this study that the carrier coating of chitosan could serve as an efficient wall material for the TiO_2_ NPs. Similarly, the surface properties of TiO_2_ NPs composite materials have been analyzed by others. Li et al. [26] found that the NPs (e.g., Ag, TiO_2_) were uniformly distributed in nano-packing film, with an irregular shape. Their results indicated that the NPs tended to improve the mechanical properties of the nano-packaging film. Yoshiki et al. [35] reported that TiO_2_ thin films have somewhat rough surfaces with micro/NPs, as observed in SEM images. According to the investigation conducted by Zhu et al. [36], The SEM image showed that the TiO_2_ NPs were uniformly incorporated in the chitosan-based coating film with irregular shapes. Conversely, Xing et al. [21], who also performed the morphological characterization of TiO_2_ nanopowders using SEM, found that the SEM image of the original particles existed in the form of agglomerates. Decreases in particle size and reductions in particle agglomeration were obtained through the use of surface modification and ultrasonication, from which it was deduced that the TiO_2_ NPs were uniformly dispersed with few agglomerates on the film. Similarly, Zhang et al. [8] performed SEM analysis on chitosan-TiO_2_ composite film, with the results showing that the TiO_2_ nano-powder was successfully and uniformly dispersed into the chitosan matrix. Therefore, it is important to investigate the effect of TiO_2_ NPs on the antimicrobial and physical properties of chitosan-based coating film.

### 3.2. AFM Analysis

In order to understand the mechanism of chitosan-based coatings with TiO_2_ NPs, the topography of the chitosan coating film was observed by AFM. The three-dimensional profiles of the coating film with different concentrations of TiO_2_ NPs are shown in Figure 4. The AFM analysis showed that the chitosan film exhibited a uniform structure, while the addition of TiO_2_ NPs changed the resultant topography of the nano-biocomposites. Thus, in the composite films with different TiO_2_ concentrations, white TiO_2_ NPs were uniformly dispersed on the film surface, while still some areas appeared darker, due to the lower local TiO_2_ NPs content. At concentrations of 0.09%, the entire analyzed surface was covered by TiO_2_ NPs, showing only a few dark areas. In addition, no large aggregates of NPs were found in any of the composite films, even at high TiO_2_ concentrations, indicating that well dispersion of NPs was realized. The NPs on the surface of the coating film with chitosan could also be validated from the SEM findings, as shown in Figure 3. The distributions of NPs on the surface of the films with different concentrations of TiO_2_ were found to be comparatively dissimilar, which could also affect the texture of the coating surface. Therefore, the parameters of surface roughness for the chitosan coating membrane were determined in terms of arithmetic mean roughness (*Ra*) and root mean square roughness (*Rq*) from the AFM height images [37]. In comparison with the chitosan membranes, the significant differences were observed in the roughness of the coating film surfaces with different concentrations of TiO_2_. As shown in Table 1, the *Ra* and *Rq* of chitosan-based coating film without TiO_2_ were 1.67 nm and 2.28 nm, respectively. After the addition of TiO_2_ NPs, the roughness of the composite films was significantly reduced. Moreover, the composite films added with different concentrations of TiO_2_ did not show significant differences between *Ra* and *Rq*. Compared with the parameters of the pure chitosan membrane, the results indicated that the appropriate amount of TiO_2_ NPs can improve the surface compactness and significantly reduce its roughness, which was consistent with the findings of Cano et al. [37].

AFM is a well-established method for characterizing topography of surfaces; however, it is recommended in previous studies as a combined approach with SEM, to ascertain the quality of homogeneity of the samples on a large scale. Moreover, the uneven distribution of peaks and valleys on the chitosan coating, significantly affected the *Ra* and *Rq* values of a film surface [38,39,40]. The *Ra* and *Rq* values of composite coatings may also be affected by the addition and interaction of NPs. Ahmad and Mirza [41] reported that the *Ra* of surfaces for nanocomposite and Pb(II) loaded nanocomposite were 48.3642 and 59.3399 nm, respectively, showing that a nanocomposite’s rough surface may be the result of the adsorption of Pb(II). Our results were consistent with an earlier study conducted by Balaji and Sethuraman [42], the *Ra* and *Rq* of a chitosan-doped-hybrid/TiO_2_ nanocomposite (5.37 and 8.63 nm) was smaller than those of undoped hybrid/TiO_2_ nanocomposite coated surfaces (9.41 and 11.80 nm), indicating that the chitosan-doped-hybrid/TiO_2_ nanocomposite had formed an adhesion. Conversely, Vijayalekshmi and Khastgir [43] found that, as an inorganic heteropolyacid content increased from 0 to 5% wt, the *Rq* of membranes increased from 4.66 to 10.7 nm, indicating that the inorganic heteropolyacid particles were well embedded in the polymer matrix. Baby Suneetha et al. also reported that increased surface roughness demonstrates that the nanocomposites may provide a large specific surface area [44]. Thus, films prepared from various solutions exhibit distinct surface properties. Furthermore, composite coatings with appropriate thicknesses and surface roughness can be applied to form protective barriers on the surface of fruits and vegetables, with the purpose of reducing their rot occurrence, improving their tissue resistance and antioxidant activity, and delaying their aging process. Tian et al. [45] found that composite coatings of chitosan/TiO_2_ NPs and chitosan/SiO_2_ NPs played an important role in defending enzyme activities and inhibiting the growth of contaminant microbes, thereby maintaining postharvest qualities and prolonging storage periods. Moreover, Meindrawan et al. [46] found that a carrageenan/ZnO NPs nanocomposite film protected mango from physical and biological damage, while the incorporation of ZnO NPs also extended its shelf life. These synergetic mechanisms still need to be further studied.

### 3.3. Thermal Gravimetric Analysis

TGA is valuable in evaluating the thermal property of coating films. The thermogravimetric (TG) and differential thermogravimetric (DTG) analyses of the composite films prepared by chitosan and different concentrations of TiO_2_ are shown in Figure 5 and Table 2. In these films, the weight loss process was roughly divided into three stages. In the first stage, the temperature rises from about 30 °C to 135 °C, and mass loss in this stage is due mainly to the evaporation of free water. The second stage involves temperatures ranging from approximately 135 °C to 450 °C, during which the rate of weight loss is rapid, with the loss of mass mainly due to the decomposition of the polymer [47,48]. In the third stage, temperatures soar from 450 °C to 800 °C, and this stage is characterized mainly by film carbonization and the decomposition of residues [18,49]. In Figure 5, the TG curve (black curve) indicates the weight loss of the composite membrane as a function of temperature, while the DTG curve (red curve) reflects the rate of weight loss of the composite membrane upon heating. As shown in Figure 5a for chitosan film only, in the first stage, the weight loss was approximately 15.5%. At the second stage, two peaks occurred in the DTG curve, at 220 °C and 280 °C, respectively, at which point the weight loss rate reached the maximum at 55.3%. At the third stage, the weight loss was 10.7%. Furthermore, as shown in Figure 5d, when 0.05% TiO_2_ was added to chitosan film, the temperature range of the three stages was 30–140 °C, 140–400 °C and 400–800 °C. The mass losses were 16.2%, 54.8% and 5.9%, respectively.

The composite films prepared with different concentrations of TiO_2_ NPs were found to exhibit better thermal stabilities. The composite film with 0.05% TiO_2_ NPs exhibited a higher decomposition temperature than those at other concentrations. TiO_2_ NPs were added into the chitosan film, which formed Ti-O bond that enhanced the interaction between chitosan molecules, and improved the thermal properties of the composite film. Qu et al. [50] reported that zein/CS/TiO_2_ films had a better thermal stability than zein/CS films. John et al. [51] also reported that the temperature of the weight loss zone increased slightly with an increase in TiO_2_ content, corresponding to the augmented thermal stability of the film. A high concentration of TiO_2_ NPs has been found to affect the activity of the molecular chain, as well as reducing relative motion, thus, not only hindering the cross-linking between different molecules, but also affecting the regularity of the network structure [50]. However, Xing et al. [52] found the addition of TiO_2_ NPs had no significant effect on the thermal stability of edible coatings and films. According to the investigation of Jbeli et al. [53], thermal analysis revealed no significant influence in the thermal stability of the material after the addition of TiO_2_ and ZnS NPs. They explained that this was caused by the similar degradation temperatures of the three systems (chitosan, CS-TiO_2_ and CS-TiO_2_/ZnS) in the second step involving decomposition. Furthermore, Morlando et al. reported that the thermostability of nanocomposites could be reduced through the addition of TiO_2_ NPs. This is probably due to the thermal conductivity of the ceramic TiO_2_ NPs, leading to an equal distribution of heat to the samples [54].

### 3.4. X-Ray Diffraction and FTIR Analysis

X-ray diffraction patterns of the original TiO_2_ NPs, pure chitosan membranes and chitosan-TiO_2_ composite membranes are shown in Figure 6. The original TiO_2_ NPs (Figure 6A) showed several characteristic peaks at 25.3°, 37.8° and 48.1°, which are consistent with the conventional peaks of anatase TiO_2_. The typical peaks of chitosan (Figure 6B(a)) appeared at 20.4°. It is evident that only one crystal form of Form Ⅱ exists in the chitosan matrix [55]. Three different forms of TiO_2_ NPs are anatase, rutile, and brookite [56]. Both chitosan and TiO_2_ diffraction peaks were observed in the composite membranes (Figure 6B(d–f)), while no other impurity peaks were found. The 2θ peaks at 25.3° confirm the TiO_2_ anatase structure without traces of the rutile and brookite phases [57,58]. However, the 2θ peak at 25.3° was not detected in the XRD spectra of the 0.01% and 0.03% samples, which may be due to the low concentration or uneven dispersion of TiO_2_ NPs in the chitosan films. FTIR spectroscopy was used to observe the interactions between the chitosan and TiO_2_ NPs. FTIR spectra of the original TiO_2_ NPs, pure chitosan membranes and chitosan-TiO_2_ composite membranes are shown in Figure 7. The FT-IR spectrum of original TiO_2_ NPs (Figure 7A) showed a very strong peak at 3440 cm^−1^, corresponding to the stretching vibration of O-H, and the bands around 2920 cm^−1^ and 2860 cm^−1^ corresponding to the C–H stretching vibration of alkyl and aldehyde groups. In the pure chitosan membranes (Figure 7B(a)), the characteristic peaks were around 3348 and 3297 cm^−1^, which were attributed to the stretching vibration of the –OH groups and –NH_2_ groups, respectively [59]. The bands at 2926 and 2879 cm^−1^ were assigned to the symmetric stretching of –CH_2_ and –CH_3_, respectively [49,60]. The chitosan films with TiO_2_ NPs (Figure 7B(b–f)) demonstrated characteristic bands at 1639 and 1563 cm^−1^ (assigned to an amide bond); 1412 cm^−1^ showed the C–N axial deformation (amine group); 1035 cm^−1^ was assigned to the stretching vibrations of C–O–C in the glycosidic linkage; while 1152 cm^−1^ was assigned to amino groups [25,61,62].

In comparison with the chitosan membranes, the chitosan peaks were found to become weak, and shift right in the XRD pattern of chitosan-TiO_2_ composite membranes. The XRD pattern of the film shows the increased intensity of the TiO_2_ peaks increasing with the amount of TiO_2_. These results may be attributable to the increasing strength of the hydrogen bonds in the chitosan complex, while complexing with TiO_2_ [59]. The above observations indicate that the incorporation of TiO_2_ into chitosan enhanced interactions between them. The physical blends versus the chemical interactions of the two components are reflected in changes in the characteristic spectra peaks [63]. Compared with the pure chitosan membrane, chitosan-TiO_2_ composite films exhibits different FT-IR spectra. The broad band between 3348 and 3297 cm^−1^ becomes slightly broader when TiO_2_ content increases, indicating that the free O–H and N–H stretching decreases, due to the hydrogen bonds interactions between the TiO_2_ molecules and –OH or –NH_2_ on chitosan chains [64]. The characteristic peak of amide I at 1639 cm^−1^ was found to shift gradually to 1644 cm^−1^ with the increase of TiO_2_ content, further confirming the formation of hydrogen bonds between TiO_2_ and chitosan. Another obvious change was a new peak emerged at 1068 cm^−1^, which was found in the FTIR spectra of chitosan-TiO_2_ composite films, and was attributed to the bond of Ti–OH [18]. The results of FTIR further suggest the interaction between chitosan and TiO_2_ NPs.

### 3.5. Effect of Chitosan-TiO_2_ Composite Films against E. coli and S. aureus

The prepared chitosan-TiO_2_ composite films show antimicrobial activity against *E. coli* and *S. aureus*, as evidenced in Figure 8, Figure 9 and Figure 10. The zones of inhibition (mm) are represented in the form of a histogram, with the mean and standard deviation are noted at the corresponding locations. As shown schematically in Figure 8, the differences were found in the values of inhibition for two kinds of the bacteria after the treatment by the chitosan-based coating films with different TiO_2_ NPs concentration, respectively. The chitosan coating without TiO_2_ showed some antibacterial activity against *E. coli* (9.86 ± 0.90 mm) and *S. aureus* (12.13 ± 0.48 mm), which are the controls for this study. The reason why pure chitosan coating films had an antimicrobial activity might be amino protonation and the subsequent cationic production, since its ultra-long molecular chain was suitable for binding *E. coli* and *S. aureus* [52]. Figure 8A shows when increasing TiO_2_ NPs concentration, the inhibition zone size of the composite coating on the *E. coli* increased compared with the control group. When the TiO_2_ NPs concentration was 0.05%, the maximum inhibition zone was found, 11.37 ± 0.76 mm, which was significantly different from the control group (*p* < 0.05), indicating that, under this concentration, the treatment showed a strong bacteriostasis effect on *E. coli*. As shown in Figure 8B, with the increase of TiO_2_ NPs concentration, the bacteriostatic zone of composite coating on *S. aureus* gradually increased. When the TiO_2_ NPs concentration was 0.09%, the inhibition zone reached a maximum of 13.55 ± 0.35 mm (*p* < 0.05).

The results showed that different concentrations of TiO_2_ NPs have been found to exert different effects on the antibacterial properties of chitosan membrane materials. In recent years, the inhibitory effects of chitosan, TiO_2_ and chitosan-TiO_2_ composite coatings on postharvest pathogens have been reported in several publications [8,21]. The antimicrobial properties of chitosan-TiO_2_ composite coatings are mainly attributed to the respective antimicrobial properties of chitosan and TiO_2_ and their synergistic effects. The amino cations contained in the molecular structure of chitosan have various biological functions, including antibacterial and anti-oxidative functions, which can act on the outer membrane of bacteria and induce sterilization. Reactive oxygen species (ROS), produced by TiO_2_ NPs, can also destroy the overall performance of the bacterial outer membrane [65]. Chitosan-TiO_2_ composite coatings act directly on the surface of bacterial cells, destroying the normal function of the cell wall (or cell membrane), and leading to the leakage of intracellular substances. In addition, chitosan-TiO_2_ composite coatings act directly on intracellular substances, and the generated oxygen free radicals (OH and O_2_^−^) attack the outer membrane, causing DNA damage, ribosome dysfunction, the interruption of electronic transport processes, as well as the oxidation or destruction of bacteria, leading to bacterial death [66]. Raut et al. [18] studied the antibacterial activity of chitosan-TiO_2_: Cu nanocomposites, SEM of *E. coli* bacterial cells showed that the nanocomposites were attached to the *E. coli* cell walls, causing direct damage and the consequent leakage of internal fluid, ultimately achieving microbial destruction. Under dark conditions, both chitosan and TiO_2_ NPs exhibited some antibacterial effects, therefore, it would be valuable to study the photo-induced antibacterial activity of composite coating materials, as well as to examine the synergistic antibacterial mechanism between the composites. Further research about the antifungal activity of composite coating materials and antibacterial activity activated with ultraviolet (UV) light is underway.

## 4. Conclusions

Chitosan-TiO_2_ coating film with different concentrations of TiO_2_ NPs were synthesized, and their physicochemical, thermal, and antimicrobial properties were systematically characterized. The TiO_2_ NPs were uniformly incorporated into chitosan-TiO_2_ coating film with an irregular shape, and a crystalline structure with the tetragonal anatase phase of TiO_2_. The chitosan-TiO_2_ coating film was found to exhibit a thermal stability superior to that of pure chitosan coating. The antibacterial properties of the composite coating material were observed to inhibit *S. aureus* more than *E. coli*. Although the composite coating material exhibited certain excellent physical and antibacterial properties, its homogeneity and transparency, antifungal properties, synergistic antibacterial mechanism, photocatalytic antibacterial property and application in the preservation of fruits and vegetables require in-depth examination, and will be the focus of further research.

## Figures and Tables

**Figure 1 nanomaterials-10-01365-f001:**
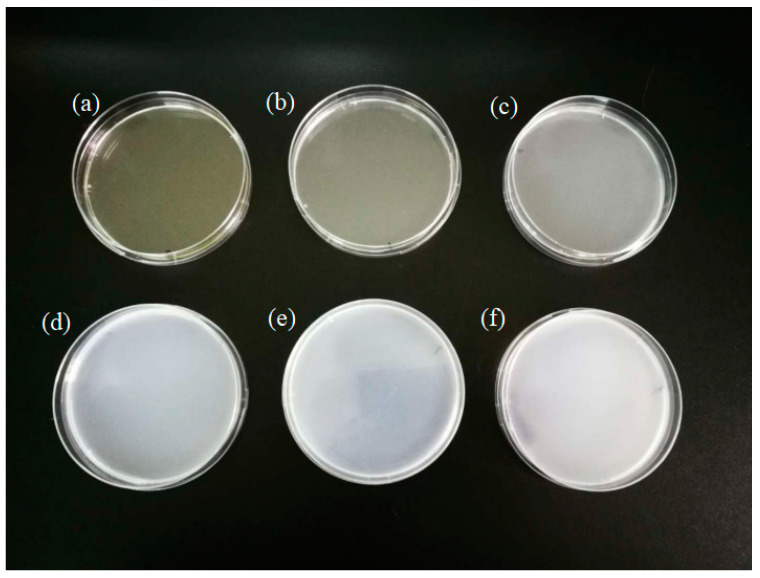
Images of chitosan-based coating film with TiO_2_ nanoparticles (**a**) 0; (**b**) 0.01%; (**c**) 0.03%; (**d**) 0.05%; (**e**) 0.07%; (**f**) 0.09%.

**Figure 2 nanomaterials-10-01365-f002:**
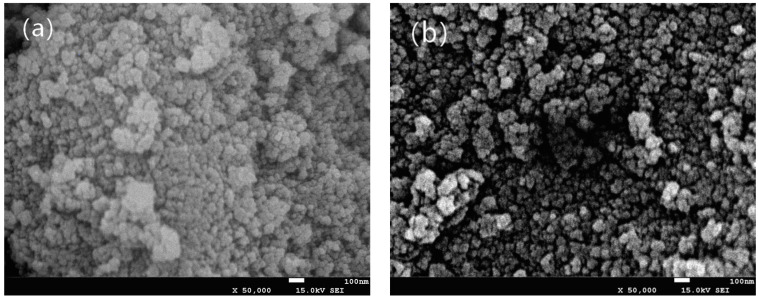
Scanning electron microscopy (SEM) images of TiO_2_ nanoparticles (**a**) and modified TiO_2_ nanoparticles (**b**).

**Figure 3 nanomaterials-10-01365-f003:**
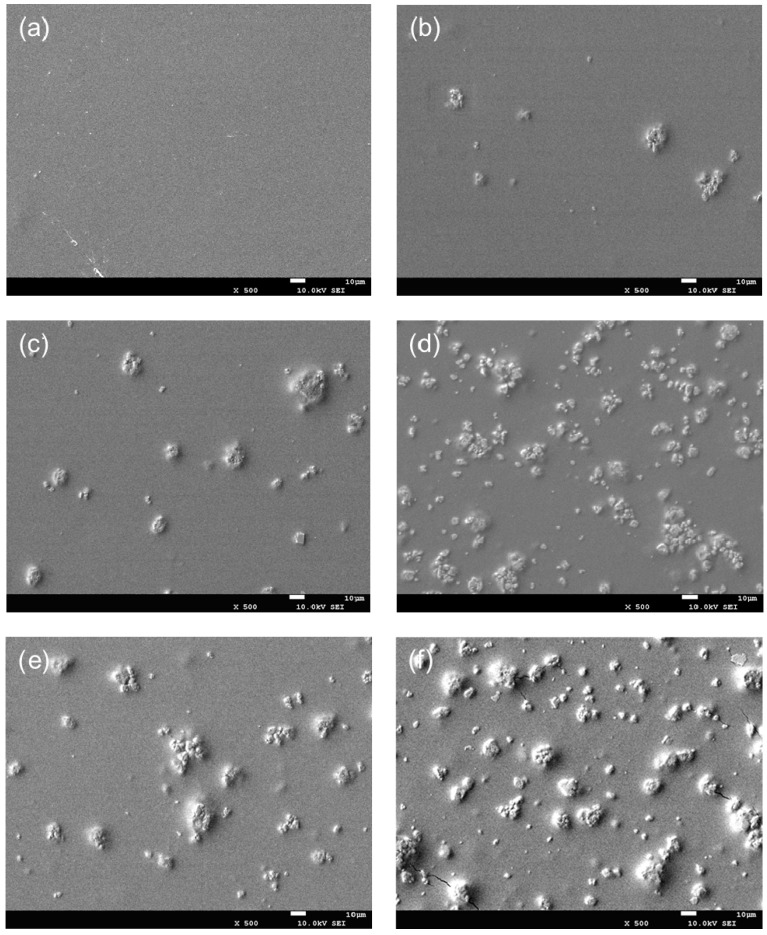
SEM images of chitosan-based coating film with TiO_2_ nanoparticles (**a**) 0; (**b**) 0.01%; (**c**) 0.03%; (**d**) 0.05%; (**e**) 0.07%; (**f**) 0.09%.

**Figure 4 nanomaterials-10-01365-f004:**
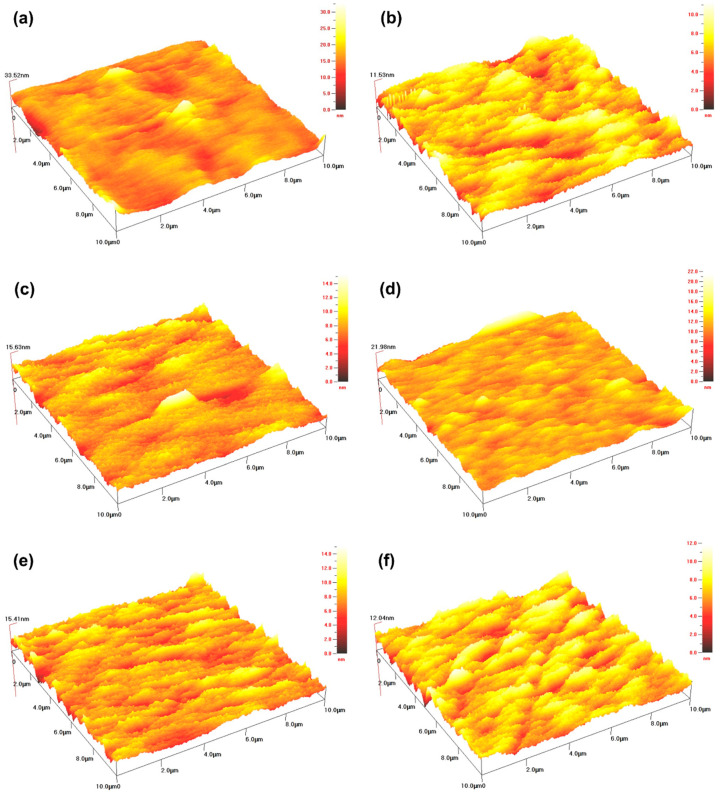
Atomic force microscopy (AFM) analysis images for the morphology of chitosan coating with TiO_2_ nanoparticles 10 μm × 10 μm; (**a**) 0; (**b**) 0.01%; (**c**) 0.03%; (**d**) 0.05%; (**e**) 0.07%; (**f**) 0.09%.

**Figure 5 nanomaterials-10-01365-f005:**
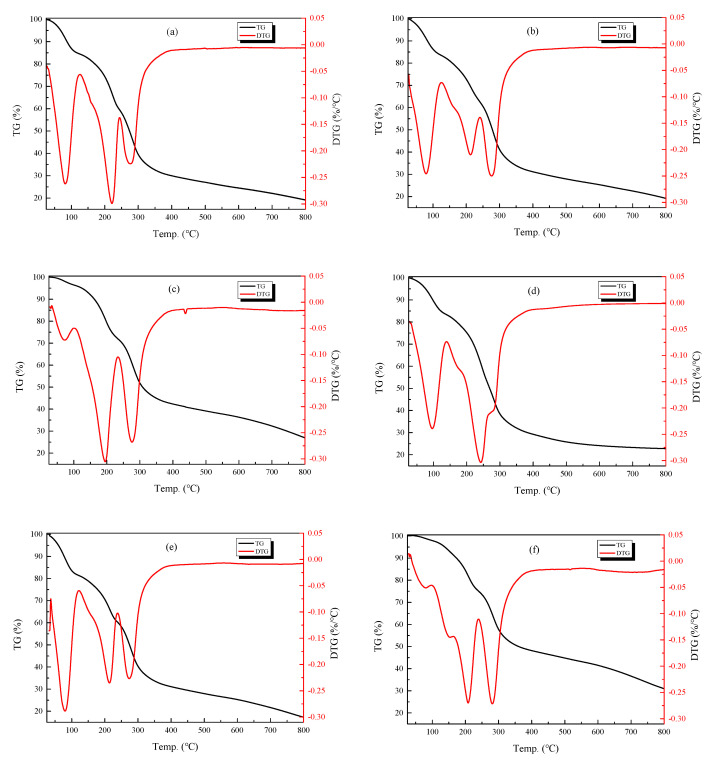
TG analysis of chitosan-based coating with different concentrations of TiO_2_ nanoparticles (**a**) 0; (**b**) 0.01%; (**c**) 0.03%; (**d**) 0.05%; (**e**) 0.07%; (**f**) 0.09%).

**Figure 6 nanomaterials-10-01365-f006:**
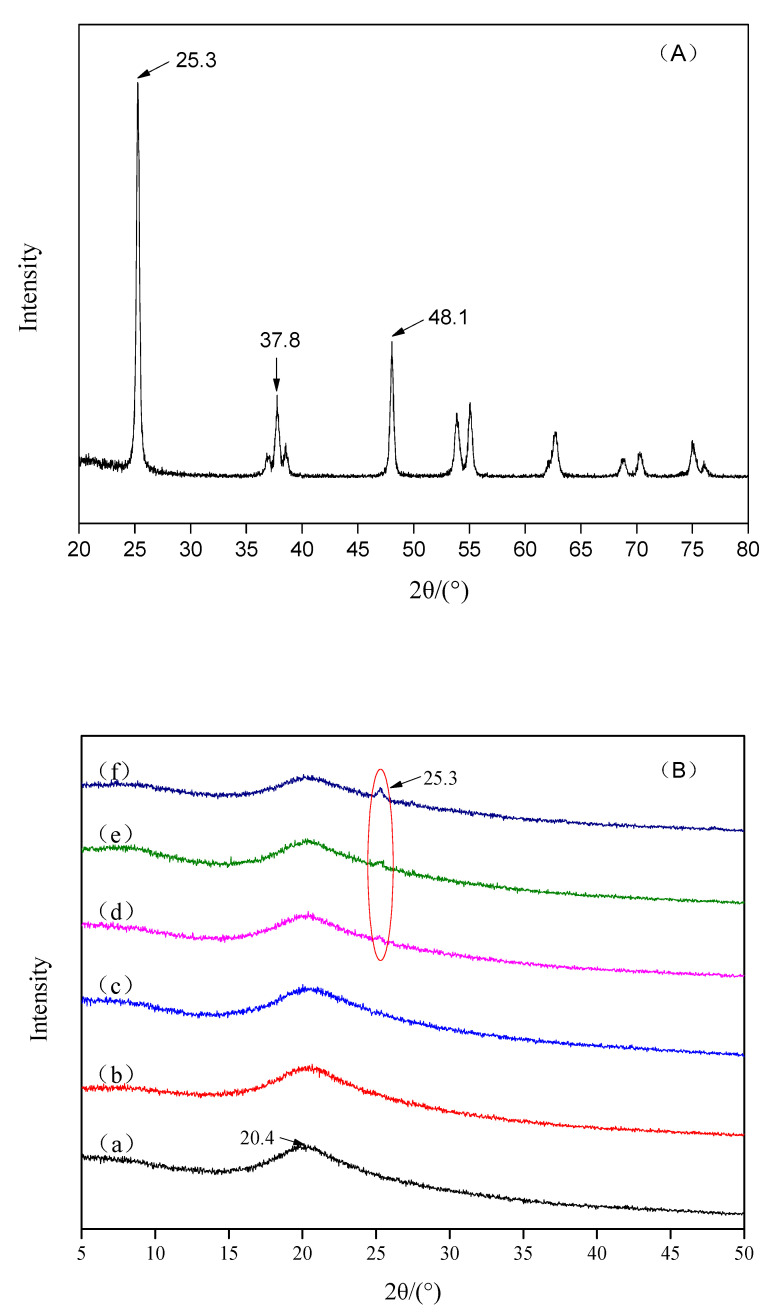
X-ray diffraction (XRD) analysis images for the morphology of original TiO_2_ nanoparticles (**A**) and chitosan coating with TiO_2_ nanoparticles (**B**) (10 μm × 10 μm; a: 0; b: 0.01%; c: 0.03%; d: 0.05%; e: 0.07%; f: 0.09%).

**Figure 7 nanomaterials-10-01365-f007:**
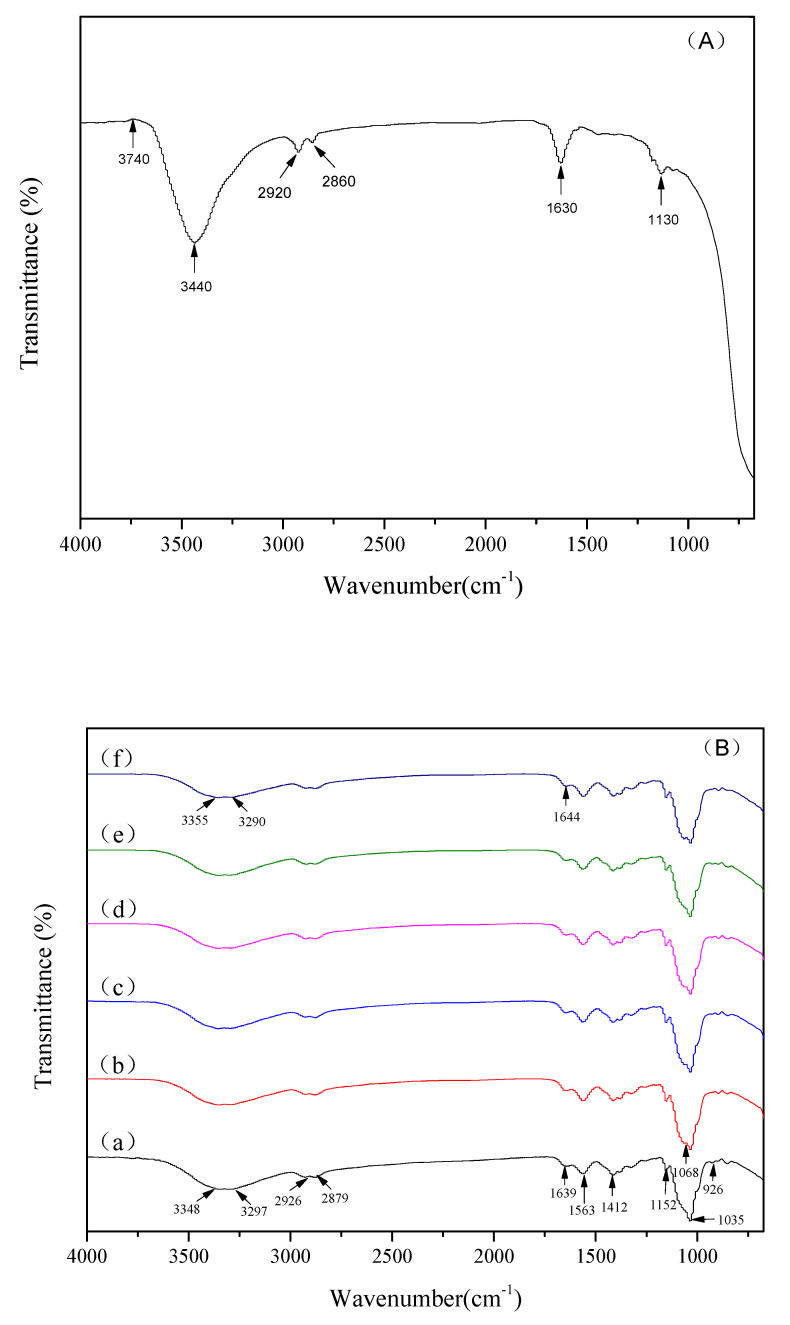
Fourier transform infrared spectroscopy (FTIR) spectra of original TiO_2_ nanoparticles (**A**) and chitosan-based coating with different concentrations of TiO_2_ nanoparticles (**B**) (a: 0; b: 0.01%; c: 0.03%; d: 0.05%; e: 0.07%; f: 0.09%).

**Figure 8 nanomaterials-10-01365-f008:**
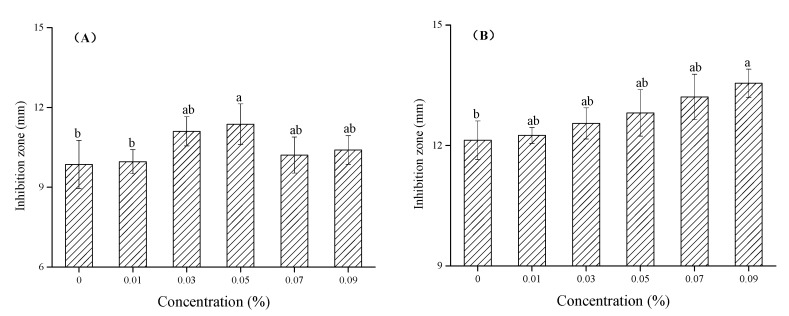
Values of inhibition for *E.coli* (**A**) and *S.aureus* (**B**) after the treatment by the chitosan-based coating film with TiO_2_ nanoparticles. Note: mean bars with different letters (a, b and ab) in the same microorganism at different concentrations of TiO_2_ nanoparticles indicate significant differences at *p* < 0.05.

**Figure 9 nanomaterials-10-01365-f009:**
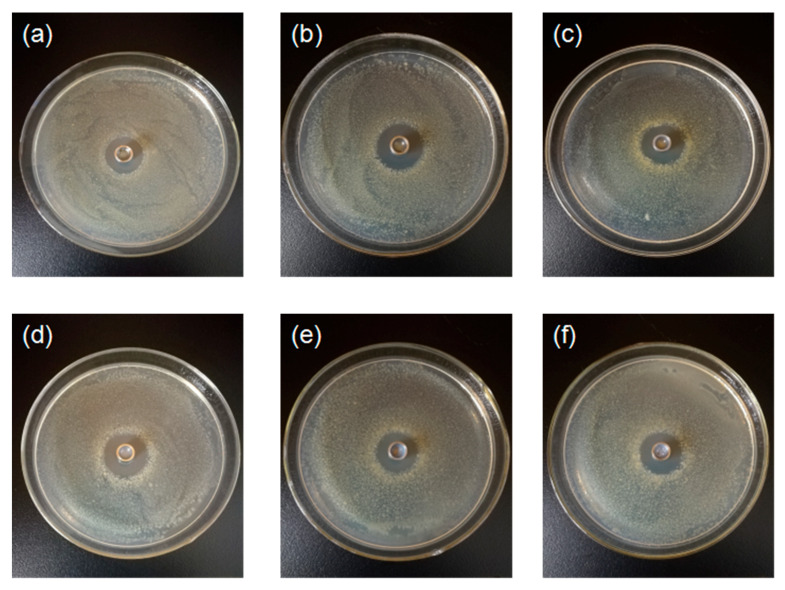
Inhibition zone for *E.coli* after the treatment by the chitosan-based coating film with TiO_2_ nanoparticles (**a**) 0; (**b**) 0.01%; (**c**) 0.03%; (**d**) 0.05%; (**e**) 0.07%; (**f**) 0.09%.

**Figure 10 nanomaterials-10-01365-f010:**
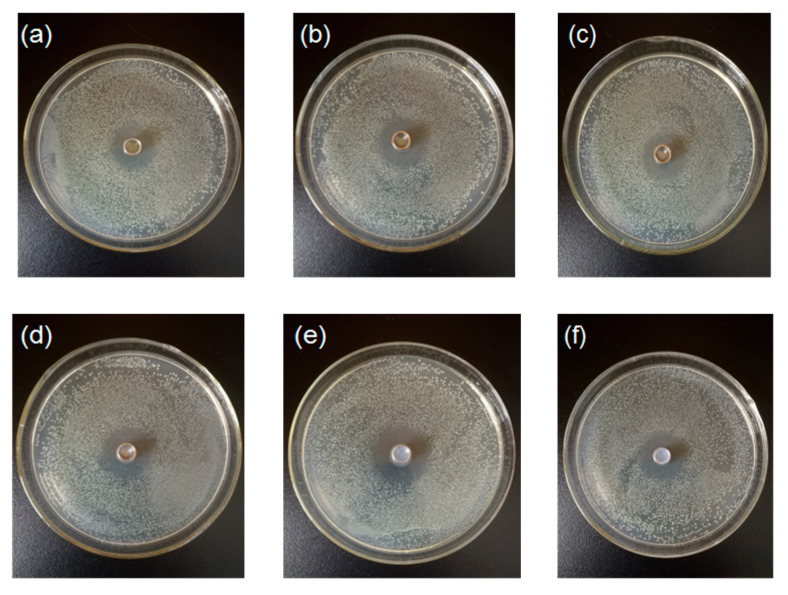
Inhibition zone for *S.aureus* after the treatment by the chitosan-based coating film with TiO_2_ nanoparticles (**a**) 0; (**b**) 0.01%; (**c**) 0.03%; (**d**) 0.05%; (**e**) 0.07%; (**f**) 0.09%.

**Table 1 nanomaterials-10-01365-t001:** Effects of TiO_2_ concentration on the arithmetic average roughness (*Ra*) and the root mean square roughness (*Rq*) of chitosan coating films (nm).

Group	*Ra* (nm)	*Rq* (nm)
Chitosan coating	1.67 ^a^ ± 0.16	2.28 ^a^ ± 0.19
Chitosan coating + 0.01 g TiO_2_	0.98 ^b^ ± 0.09	1.32 ^b^ ± 0.02
Chitosan coating + 0.03 g TiO_2_	1.02 ^b^ ± 0.08	1.28 ^b^ ± 0.07
Chitosan coating + 0.05g TiO_2_	1.04 ^b^ ± 0.08	1.31 ^b^ ± 0.14
Chitosan coating + 0.07 g TiO_2_	1.22 ^b^ ± 0.23	1.58 ^b^ ± 0.31
Chitosan coating + 0.09 g TiO_2_	1.06 ^b^ ± 0.05	1.34 ^b^ ± 0.07

Note: Each data represents the mean value ± SD. Different letters (a and b) within columns indicate significant differences at *p* < 0.05.

**Table 2 nanomaterials-10-01365-t002:** Values of thermogravimetric (TG) analysis after the treatment by the chitosan-based coating film with TiO_2_ nanoparticles.

Group	The First Stage	The Second Stage		The Third Stage
	TG (%)	TG (%)	DTG	TG (%)
Chitosan coating	15.5	55.3	220 °C, 280 °C	10.7
Chitosan coating + 0.01 g TiO_2_	16.2	51.8	215 °C, 280 °C	13.3
Chitosan coating + 0.03 g TiO_2_	4.7	37.8	196 °C, 281 °C	15.7
Chitosan coating + 0.05 g TiO_2_	16.2	54.8	246 °C, 283 °C	5.9
Chitosan coating + 0.07 g TiO_2_	18.6	50.1	216 °C, 280 °C	14.9
Chitosan coating + 0.09 g TiO_2_	8.2	46.2	209 °C, 284 °C	15.8

Note: Values of DTG represent the peaks of the second stage in the DTG curve.

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
