# Peer review of "Effects of Different TiO_2_ Nanoparticles Concentrations on the Physical and Antibacterial Activities of Chitosan-Based Coating Film"

_nanomaterials, 2020, doi:10.3390/nano10071365_

Round 1

Reviewer 1 Report

The proposed manuscript is interesting and I suggest its publication after some minor corrections:

  1. Experimental part:

- explain why sodium laurate was added in the preparation of surface-modified TiO2 nanoparticles;

- explain why glycerol was added in the preparation of chitosan-based coating film with modified TiO2 nanoparticles.

  1. Thermal gravimetric analysis: The discussion of this part is too wordy at page 10. I would suggest to summarize the TG and DTG in a table and expose only the characteristics of the best sample in comparison to the pure chitosan coating film.
  2. XRD analysis (Figure 6): Explain why 2θ peak at 25.3° was not detected in the XRD spectra of the 0.01% and 0.03% samples.
  3. Antibacterial activity (Figure 8 and Figures 9 and 10): Explain the formation of the inhibition zone in the case of pure chitosan coating film, if it is well known that chitosan acts according to the bio-barrier formation mechanism and does no leach out. It only kills microorganisms that comes in direct contact.

Author Response

Manuscript ID: nanomaterials-834698

Title: Effects of different TiO2 nanoparticles concentrations on the physical and antibacterial activities of chitosan-based coating film

Dear reviewer,

The changes to the manuscript can be got form the red colored text. The responses for the comments of the reviewer are shown as follow.

We appreciate valuable and constructive comments from reviewer on our manuscript.

Response to the suggestions of reviewer:

  1. Experimental part:

- explain why sodium laurate was added in the preparation of surface-modified TiO2 nanoparticles;

The explanation has been revised as follow.

Sodium laurate could expose non-polar groups and disperse TiO2 nanoparticles.

- explain why glycerol was added in the preparation of chitosan-based coating film with modified TiO2 nanoparticles.

The explanation has been revised as follow.

Adding glycerin could improve the mechanical properties of the composite film.

  1. Thermal gravimetric analysis: The discussion of this part is too wordy at page 10. I would suggest to summarize the TG and DTG in a table and expose only the characteristics of the best sample in comparison to the pure chitosan coating film.

The partial discussion has been deleted in the manuscript and the table has been revised as follow.

Table 2 Values of TG analysis after the treatment by the chitosan-based coating film with TiO2 nanoparticles. 

Note: Values of DTG represent the peaks of the second stage in the DTG curve.

Group

the first stage

The second stage

the third stage

TG (%)

TG (%)

DTG

TG (%)

Chitosan coating

15.5

55.3

220°C, 280°C

107

Chitosan coating+0.01g TiO2

16.2

51.8

215°C, 280°C

13.3

Chitosan coating+0.03g TiO2

4.7

37.8

196°C, 281°C

15.7

Chitosan coating+0.05g TiO2

16.2

54.8

246°C, 283°C

5.9

Chitosan coating+0.07g TiO2

18.6

50.1

216°C, 280°C

14.9

Chitosan coating+0.09g TiO2

8.2

46.2

209°C, 284°C

15.8

  1. XRD analysis (Figure 6): Explain why 2θ peak at 25.3° was not detected in the XRD spectra of the 0.01% and 0.03% samples.

The explanation has been revised as follow.

However, 2θ peak at 25.3° was not detected in the XRD spectra of the 0.01% and 0.03% samples, which may be due to the low concentration or uneven dispersion of TiO2 nanoparticles.

  1. Antibacterial activity (Figure 8 and Figures 9 and 10): Explain the formation of the inhibition zone in the case of pure chitosan coating film, if it is well known that chitosan acts according to the bio-barrier formation mechanism and does no leach out. It only kills microorganisms that comes in direct contact.

The explanation has been revised as follow.

The renson why pure chitosan coating film had antimicrobial activity might be amino protonation and the subsequent cationic production, since its ultra-long molecular chain was suitable for binding E. coli and S. aureus [54].

Very thanks for the valuable and constructive comments from reviewer on our manuscript.

Best wishes and happy every day.

Yage Xing

Reviewer 2 Report

The manuscript entitled “Effects of different TiO2 nanoparticles concentrations on the physical and antibacterial activities of chitosan-based coating film” by Xing et al., investigates the effect of different concentrations of titanium dioxide (TiO2) nanoparticles on the structure and antimicrobial activity of chitosan-based coating films. The chitosan-TiO2 nanocomposite exhibited an inhibitory effect on the growth of Escherichia coli and Staphylococcus aureus, representing a potential packaging system for prolonging the shelf-life of fruits and vegetables. The manuscript is scientifically sound. Its structure and organization, the methodology, the combined results and discussion section and the main conclusions and writing are correct. The experimental protocols are correctly performed. However, few points must be revised.

Minor points:

-Paragraph 2.2., line 101, change “Five grams of TiO2 nanoparticles was gently dispersed” with “TiO2 nanoparticles (5 g) were gently dispersed”.

-Paragraph 2.2., line 102, change “mol/L” with “M” to indicate the molarity.

-Paragraph 2.2., line 105, the rpm of centrifuge must be converted in acceleration relative to ‘g’ (relative centrifugal force, RCF) or the type of centrifuge used must be indicated.

- Paragraph 2.3., line 109, change “One gram of chitosan powder was dissolved” in “Chitosan powder (1 g) was dissolved”.

- Paragraph 2.3., line 119, change “One gram of chitosan powder was then dissolved” in “Chitosan powder (1 g) was then dissolved”.

-In Figure 8, the values at the top of each bar are superfluous since they are already reported in the main text. Please revise both the caption and the figure by deleting the values.

Author Response

Manuscript ID: nanomaterials-834698

Title: Effects of different TiO2 nanoparticles concentrations on the physical and antibacterial activities of chitosan-based coating film

Dear reviewer,

The changes to the manuscript can be got form the red colored text. The responses for the comments of the reviewer are shown as follow.

We appreciate valuable and constructive comments from reviewer on our manuscript.

Response to the suggestions of reviewer:

-Paragraph 2.2., line 101, change “Five grams of TiO2 nanoparticles was gently dispersed” with “TiO2 nanoparticles (5 g) were gently dispersed”.

The sentences have been revised in the manuscript.

-Paragraph 2.2., line 102, change “mol/L” with “M” to indicate the molarity.

The sentences have been revised in the manuscript.

-Paragraph 2.2., line 105, the rpm of centrifuge must be converted in acceleration relative to ‘g’ (relative centrifugal force, RCF) or the type of centrifuge used must be indicated.

The rpm of centrifuge have been revised as follow.

12100 g

- Paragraph 2.3., line 109, change “One gram of chitosan powder was dissolved” in “Chitosan powder (1 g) was dissolved”.

The sentences have been revised in the manuscript.

- Paragraph 2.3., line 119, change “One gram of chitosan powder was then dissolved” in “Chitosan powder (1 g) was then dissolved”.

The sentences have been revised in the manuscript.

-In Figure 8, the values at the top of each bar are superfluous since they are already reported in the main text. Please revise both the caption and the figure by deleting the values.

The Figure 8 have been revised as follow.

Very thanks for the valuable and constructive comments from reviewer on our manuscript.

Best wishes and happy every day.

Yage Xing
